# Cell atlas of the regenerating human liver after portal vein embolization

Agnieska Brazovskaja [1,10], Tomás Gomes[2,10] ✉, Rene Holtackers[2], Philipp Wahle [2], Christiane Körner [3], Zhisong He [2], Theresa Schaffer [1], Julian Connor Eckel [3], René Hänsel [3,4], Malgorzata Santel[2], Makiko Seimiya[2], Timm Denecke[5], Michael Dannemann [1,6], Mario Brosch[7,8], Jochen Hampe [7,8], Daniel Seehofer [3], Georg Damm [3,11] ✉, J. Gray Camp[1,9,11] ✉ & Barbara Treutlein [1,2,11] ✉

The liver has the remarkable capacity to regenerate. In the clinic, regeneration is induced by portal vein embolization, which redirects portal blood flow, resulting in liver hypertrophy in locations with increased blood supply, and atrophy of embolized segments. Here, we apply single-cell and single-nucleus transcriptomics on healthy, hypertrophied, and atrophied patient-derived liver samples to explore cell states in the regenerating liver. Our data unveils pervasive upregulation of genes associated with developmental processes, cellular adhesion, and inflammation in post-portal vein embolization liver, disrupted portal-central hepatocyte zonation, and altered cell subtype composition of endothelial and immune cells. Interlineage crosstalk analysis reveals mesenchymal cells as an interaction hub between immune and endothelial cells, and highlights the importance of extracellular matrix proteins in liver regeneration. Moreover, we establish tissue-scale iterative indirect immunofluorescence imaging for high-dimensional spatial analysis of perivascular microenvironments, uncovering changes to tissue architecture in regenerating liver lobules. Altogether, our data is a rich resource revealing cellular and histological changes in human liver regeneration.

The human liver is capable of executing diverse metabolic, immunological and detoxification functions, by relying on various cell types organized into hexagonal lobules and following patterns of blood flow[1,2]. Lobules are structured by blood vessels forming portal triads, and converging at the center into the central vein. This portal-to-central axis correlates with phenotypic and metabolic zonation of hepatocytes and liver sinusoidal endothelial cells (LSECs)[3–8]. Recent single-cell RNA-sequencing (scRNA-seq) studies have shed light into liver biology in homeostasis[9–12] and disease[13–15], with unprecedented resolution of cell type heterogeneity, tissue structure, intercellular contacts, and reconstruction of hepatocyte and LSEC zonation. Despite

[1]Max Planck Institute for Evolutionary Anthropology, Leipzig, Germany. [2]Department of Biosystems Science and Engineering, ETH Zürich, Basel, Switzerland. [3]Department of Hepatobiliary Surgery and Visceral Transplantation, University Hospital, Leipzig University, Leipzig, Germany. [4]Institute for Medical Informatics, Statistics and Epidemiology (IMISE), Leipzig University, Leipzig, Germany. [5]Department of Diagnostic and Interventional Radiology, Leipzig University, Leipzig, Germany. [6]Institute of Genomics, University of Tartu, Tartu, Estonia. [7]Medical Department 1, University Hospital Dresden, Technical University Dresden, Dresden, Germany. [8]Center for Regenerative Therapies Dresden (CRTD), Technical University Dresden, Dresden, Germany. [9]Institute of Human Biology (IHB), Roche Pharma Research and Early Development, Roche Innovation Center Basel, Basel, Switzerland. [10]These authors contributed equally: Agnieska Brazovskaja, Tomás Gomes. [11]These authors jointly supervised this work: Georg Damm, J. Gray Camp, Barbara Treutlein. ✉e-mail: tomas.gomes@medicina.ulisboa.pt; georg.damm@medizin.uni-leipzig.de; gray.camp@roche.com; barbara.treutlein@bsse.ethz.ch

these efforts, obtaining a representative sampling of all cell types in this tissue is still the object of intense research[16,17].

The human liver is able to restore itself and expand when injured or partially removed[18]. Liver regeneration comprises hypertrophy (increase in cell size) and hyperplasia (increase in cell number) as compensation for lost tissue mass[19]. The cellular and molecular pathways underlying liver regeneration have been extensively studied in non-human models[20] using experimental approaches such as partial hepatectomy or pharmacologically-induced liver damage[21]. Due to these being unachievable in humans, a knowledge gap exists between animal and clinical studies on liver regeneration.

Hepatic resection has been used in the clinic to treat different liver diseases and opened opportunities to work on transplantation strategies[22]. Portal vein embolization (PVE) is a technique applied to avoid liver insufficiency in patients undergoing liver resection. It redirects blood flow to specific liver segments, resulting in a hypertrophic regenerated portion that functionally compensates the atrophied embolized section to be surgically removed[22,23].

Here we apply single-cell and single-nucleus transcriptomics on healthy, hypertrophied (regenerating), and atrophied (embolized) human liver biopsies to explore cell states within this paradigm of human liver regeneration. The data reveal changes to cell type proportions, hepatocyte zonation, and intercellular communication in the affected human liver. We further establish highly multiplexed immunohistochemistry on human liver tissue sections, validating the coupling of cellular and histological changes post-PVE. Together, these data unravel transcriptomic, cellular, and histological aspects of human liver regeneration.

## Results

### Identifying cell types in fresh and frozen liver samples

We performed single-cell and single-nucleus RNA-seq to generate a map of transcriptional profiles of the healthy human liver. Human liver samples were acquired from patients undergoing liver resection due to benign liver diseases (see Methods, Supplementary Data 1) and were graded as unobtrusive by pathologists with slight signs of steatosis in only one case. We developed downstream processing protocols in two different ways (Fig. 1a). First, on surgery day we obtained biopsies for an immediate experiment (referred to as fresh), and established a workflow to generate single-cell suspensions for parenchymal (hepatocytes) and non-parenchymal (cholangiocytes, endothelial, immune and mesenchymal cells) fractions (Supplementary Fig. 1a). Second, we used liquid nitrogen to snap freeze a portion of the biopsy, which we were able to store long term and use for preparation of single-nucleus suspensions (referred to as frozen) (Supplementary Fig. 2a, b). For frozen samples, single-nuclei were liberated from the tissue using Dounce homogenization and isolated using fluorescence-activated cell sorting (FACS) with no additional fractionation as was performed in the fresh samples.

We generated and analyzed transcriptomes of ~21,000 cells and ~9400 nuclei from three fresh and three frozen samples, with two samples in each condition derived from the same donor (see Methods, Supplementary Data 1). We analyzed fresh or frozen tissue datasets separately after integrating the cells from different donors[24,25] (see Methods), and visualized cell heterogeneity using a uniform manifold approximation and projection (UMAP) embedding (Fig. 1b). Clustering revealed 5 major cell populations[9,12] in both fresh and frozen tissue datasets representing hepatocyte, cholangiocyte (bile duct epithelial cells), endothelium, mesenchyme, and immune cells (Fig. 1b). Despite the lower number of genes detected per cell population in the frozen samples, cell type marker genes were consistently detected, and differentially expressed (DE) genes between cell types were strongly correlated between fresh and frozen tissue datasets (Fig. 1c–e, Supplementary Figs. 1, 2). The higher depth and sensitivity of the data from fresh whole-cell samples enabled a better resolution of cell

subpopulations (Supplementary Fig. 1e, f). However, the unfractionated sampling using snRNA-seq provided a more representative survey of certain cell populations. For example, the proportion of mesenchymal cells in the frozen tissue dataset was over thirty times higher compared to fresh tissue data (Supplementary Data 2, 3). In addition, the enrichment for non-parenchymal cells when processing fresh samples revealed a larger fraction of immune cells and a more diverse set of subpopulations (Supplementary Fig. 1). We were able to identify previously described zonation patterns in hepatocytes (Supplementary Fig. 3) and in LSECs (Supplementary Fig. 4) in fresh and frozen datasets, with expression signatures shared between the experimental setups. Altogether, we established strategies for generating single-cell and single-nucleus transcriptome atlases from fresh and frozen patient-derived liver samples.

### Human liver transcriptional landscape after PVE

To investigate cellular processes specific to the regenerating human liver we obtained freshly resected liver tissue samples from six patients who underwent a preoperative medical procedure called PVE (see Methods). During PVE, the portal vein branching to the diseased part of the liver is blocked, or embolized with metal coils, and the future liver remnant receiving increased portal blood flow expands over time (Fig. 2a, b). On resection day, we received two tissue samples from the same donor that we refer to as regenerating and embolized samples. All samples were taken with safety distance to any malign lesions by the senior surgeon in close consultation with the department for pathology (see Methods, Supplementary Data 1). We isolated parenchymal and non-parenchymal cell fractions as described above for the healthy condition, and performed scRNA-seq on each fraction. We integrated expression data from all three conditions and projected the data using UMAP (Fig. 2c, see Methods). In the combined dataset we identified hepatocytes, endothelial cells (EC), cholangiocytes, immune cells, and mesenchymal cells as the major cell types (Fig. 2d and Supplementary Fig. 5), recovered in similar proportions from all donors and conditions (Fig. 2e and Supplementary Fig. 5c).

For each cell type we compared gene expression levels in post-PVE conditions to their counterparts in healthy liver tissue. DE genes in all cell types except hepatocytes were more often upregulated in both regenerating and embolized conditions (Fig. 2f, Supplementary Data 4). Such regeneration kinetics are consistent with previous results showing that hepatocytes are the first cell types entering the regeneration program after hepatectomy, and are followed by cell types responsible for tissue reorganization, including liver vasculature and bile ducts reestablishment[26,27]. Analysis of DE genes within cholangiocytes (Fig. 2g), ECs (Fig. 2h), hepatocytes (Fig. 2i), and immune cells (Fig. 2j), showed upregulated genes in both regenerating and embolized tissue, as well as features specific to each condition, compared to the healthy reference.

Mesenchymal cells as a whole did not present any DE genes between conditions, likely due to their low cell numbers (23, 104, and 67 in healthy, embolized and regenerating, respectively). Nonetheless, they revealed some heterogeneity, dividing into fibroblasts, hepatic stellate cells (HSC), and vascular smooth muscle cells (VSMC) (Supplementary Fig. 6a–d), based on previously published gene signatures[28] (Supplementary Fig. 6e). Despite the low number of cells recovered, an increased proportion of fibroblasts and HSC was observed in post-PVE samples, particularly in the regenerating liver (Supplementary Fig. 6f, g), indicating a potential role for these cells in human liver regeneration.

DE genes revealed an enrichment for multiple expression programs in post-PVE samples (Fig. 2k–n). Gene ontology (GO) Terms related to development ("Cell morphogenesis" in cholangiocytes and "Vasculature development" in ECs) were present both in regenerating and embolized data, with a stronger enrichment result in the former. Supporting this difference in gene expression, histological analysis of

PVE tissues showed several cholangiocytes and endothelial cell features that differed from healthy liver tissue. Embolized tissue showed a high number of various undefined angiogenic events within the liver lobule parenchyma. Regenerating tissues showed portal fields with a higher number of bile ducts compared to healthy controls, as well as locations within the lobular parenchyma containing biliary and vascular cells, suggesting remodeling of portal fields and newly formed portal structures as a precursor for portal fields, respectively (Supplementary Fig. 7). GO terms associated with cellular adhesion pervade all major cell types, attesting to the considerable morphogenic changes induced by PVE. ECs and hepatocytes also revealed an upregulation of innate immune and inflammation programs (Fig. 2l, m),

pathways with a critical role in initiating the regenerative process[29]. We also noted a specific enrichment for various pathways in immune cells in the regenerating tissue (Fig. 2n), an indication of their pivotal role in the process.

Among the upregulated transcripts in hepatocytes from regenerating and embolized tissues were also genes known to have a zone-specific expression (*HAMP, CRP, IGFBP2, SAA1, SAA2*; Fig. 2i, Supplementary Data 4), potentially indicating gene expression alterations in zonation.

## Periportal-like hepatocytes predominate in post-PVE liver

We explored whether gene expression in post-PVE hepatocytes showed a similar zonation to healthy tissue (Fig. 3a, b, Supplementary

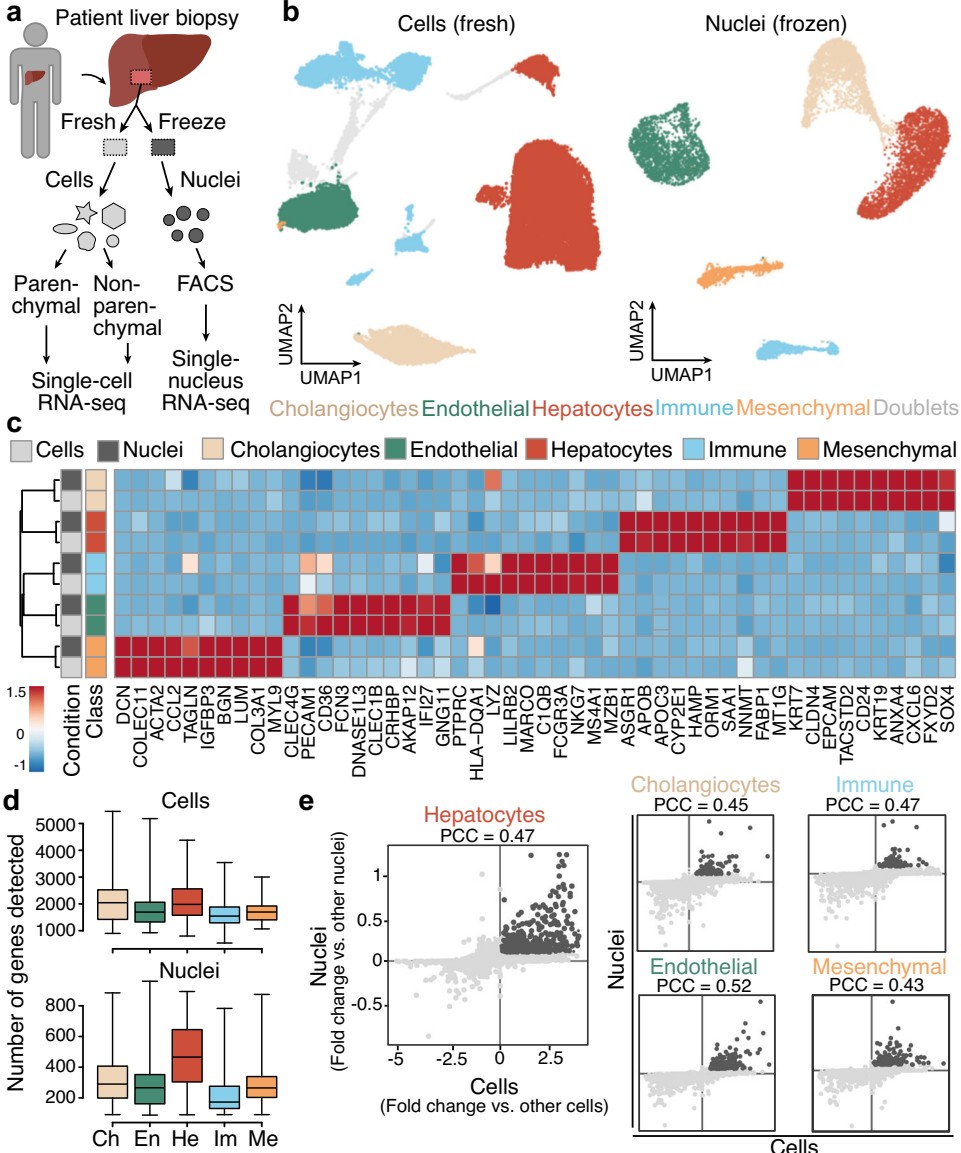

**Fig. 1 | Fresh single-cell and frozen single-nucleus RNA-seq reveal major cell populations in adult healthy liver specimens. a** Single-cell and single-nucleus RNA-seq experiments were performed on patient-derived fresh (*n* = 3) and frozen (*n* = 3) liver tissues. Prior to single-cell RNA-seq, cells isolated from the fresh tissue were partitioned into parenchymal (hepatocytes) and non-parenchymal (other hepatic cell types) fractions (left). Prior to single-nucleus RNA-seq, nuclei from snap-frozen tissues were isolated using fluorescence activated cell sorting (right). **b** UMAP plot of transcriptomes from fresh (left) and frozen (right) tissue datasets colored by major cell type (see Supplementary Figs. 1, 2 for each donor's representation). Gray represents potential doublets. **c** Scaled expression of marker genes

per major cell type across fresh and frozen tissue data (cells and nuclei are represented in rows, genes in columns). **d** Number of detected genes per cell (top) and nucleus (bottom) across major cell types. Ch cholangiocytes, En endothelial cells, He hepatocytes, Im immune cells, Me mesenchymal cells. Boxplot shows the median (center line), 25th, and 75th percentile (lower and upper boundary), and the whiskers indicate the minimum and maximum values. **e** Comparison of marker gene detection is shown for each major cell type in fresh and frozen samples (dark gray represents genes shared between both experimental setups; PCC, Pearson correlation coefficient). Source data are provided as a Source Data file for (**b–e**).

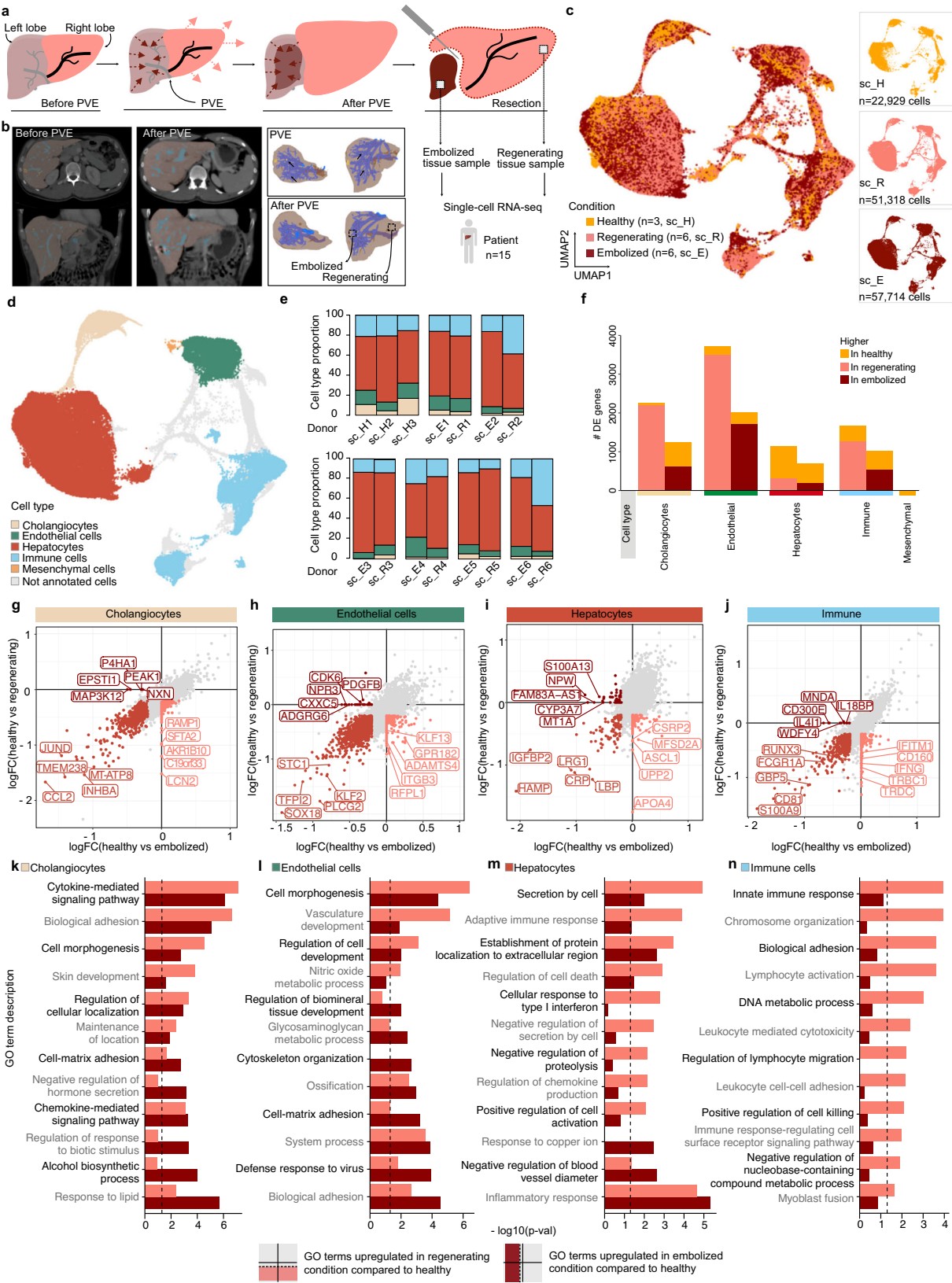

Fig. 3, Supplementary Fig. 8a, b). UMAP embeddings of post-PVE hepatocytes revealed a portal-to-central signature gradient similar to healthy hepatocytes (Fig. 3c). Unsupervised projection of hepatocytes into a one-dimensional axis (see Methods) ordered healthy cells according to liver lobule zonation, as attested in all three conditions by the expression profiles of known zonation markers (Fig. 3d). A subset

of genes were found to have differential zonation between healthy and regenerating or embolized samples (Supplementary Fig. 8c, e, Supplementary Data 5). In both PVE conditions, genes that differed from healthy were associated with cellular respiration (e.g., *NDUFA3, SOD2* in regenerating; *CP, MT-ND6* in embolized) and lipid metabolism (e.g., *APOA5* in regenerating; *APOC3, APOC2* in embolized). In regenerating

**Fig. 2 | Transcriptional landscape of human liver cells after portal vein embolization (PVE). a** Schematic shows the portal vein embolization (PVE) procedure and tissue sampling for the scRNA-seq experiments. Portal vein branching toward diseased liver tissue (left lobe) prior to resection is blocked or embolized. Adjacent tissue with redirected blood flow (right lobe) expands over time. ScRNA-seq is performed on two samples derived from regenerating and embolized liver tissues on the day of liver resection. **b** Computed tomography (CT) scans of pre-PVE (left) and post-PVE (middle) livers are shown along with 3D tissue reconstructions (right), respectively. UMAP plot of merged healthy (*n* = 3), regenerating (*n* = 6) and embolized (*n* = 6) fresh liver samples, colored by condition (**c**) and major cell type

(**d**). **e** Major cell type proportion per donor and condition. **f** Number of differentially expressed genes between healthy and regenerating or embolized samples per major cell type. Gene expression log fold change for each PVE condition compared to healthy (x-axis: embolized; y-axis: regenerating) for cholangiocytes (**g**), Endothelial cells (**h**), Hepatocytes (**i**), and Immune cells (**j**). **k–n**, Enriched gene ontology terms for DE genes per major cell type, comparing regenerating or embolized to healthy. Top 6 terms were selected for each condition (two-sided hypergeometric test, dashed line shows Benjamini-Hochberg adjusted *p* = 0.05). Source data are provided as a Source Data file for (**c**, **d–n**).

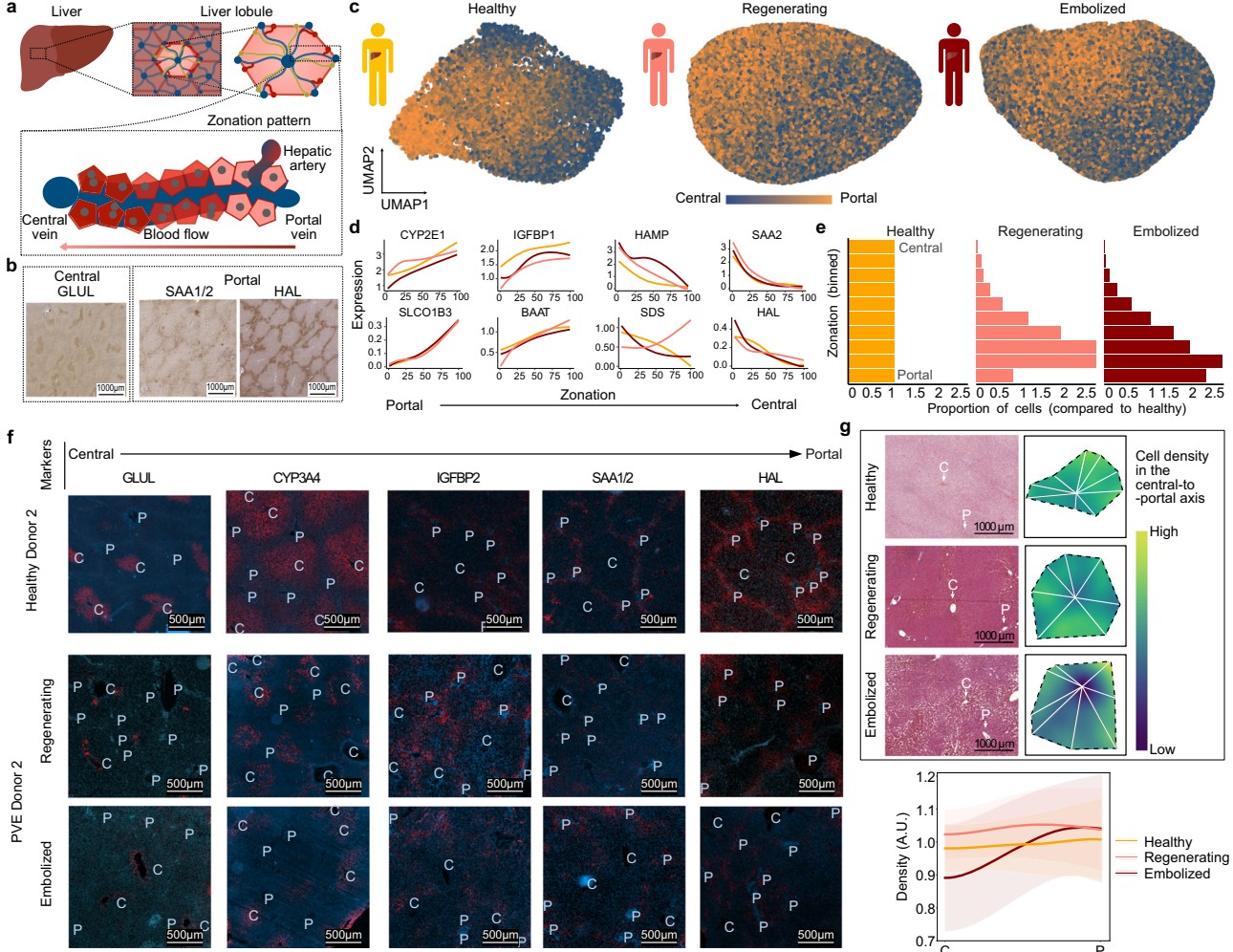

**Fig. 3 | Spatial zonation patterns are altered in regenerating and embolized tissue hepatocytes. a** Schematic illustrating the structure of a liver lobule. **b** 3,3'-Diaminobenzidine (DAB) stainings of central and portal zone-specific protein expression within healthy tissue hepatocytes. **c** Gene expression signature of zonation within healthy (right), regenerating (middle), and embolized (left) tissue hepatocytes. Cells in UMAP plots are colored based on the cumulative expression of central (blue) or portal (orange) zonation marker genes. **d** Pseudozonation expression patterns of representative zonation marker genes for each medical

condition. **e** Histograms showing the proportion of hepatocytes in zonation bins across conditions. **f** Immunofluorescence tissue staining for liver tissue for various hepatocyte zonation markers. P portal vessel, C central vessel. See also Supplementary Figs. 9, 10. **g**, Top: Representative H/E stainings showing liver lobule segmentation to measure cell density along a central-to-portal axis; Bottom: Mean ± s.e. for cell density along the central-to-portal axis in each condition. Source data are provided as a Source Data file for (**d**, **e**).

hepatocytes, we also observed genes involved in response to toxic substances (*CYP2A6, GSTO1, MT2A, CYB5A*, and *ASS1*). These data suggest that metabolic homeostasis may be disrupted or in flux after the PVE procedure.

Projection of post-PVE hepatocytes into the healthy pseudozonation reference also indicated a depletion of cells with a pericentral expression signature and a corresponding enrichment in cells with a midzonal and periportal-like signature (Fig. 3e). To understand if these

gene expression changes may reflect alterations to the lobule spatial organization, we examined the presence of known hepatocyte zonation markers in various liver lobules through immunofluorescence (Fig. 3f, Supplementary Fig. 9) and H/E stainings (Supplementary Fig. 10). These revealed liver lobules with less clear borders and more disorganized tissue architecture in PVE conditions, although markers were still broadly associated with their known portal/central location, as expected from the scRNA-seq results (Fig. 3d). The central marker

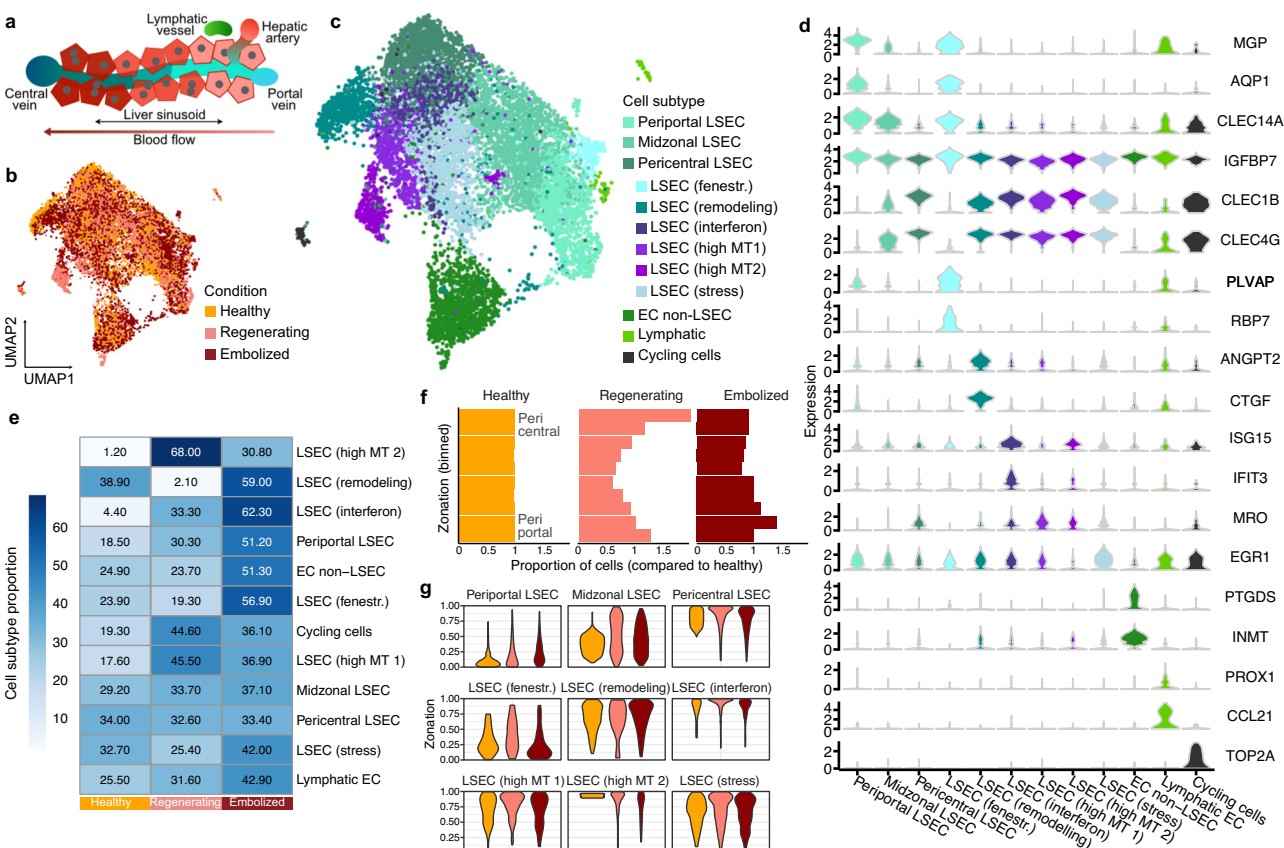

**Fig. 4 | Liver endothelial cell heterogeneity and inferred zonation after PVE.**
**a** Schematic illustrates diversity of endothelial cells along the portal-central axis within a liver lobule. UMAP plots representing combined ECs from all medical conditions are colored by condition (**b**) or annotated subpopulation (**c**) (EC endothelial cells, LSECs liver sinusoidal endothelial cells, fenestr. fenestrated, MT mitochondrial). **d** Violin plots show distribution of normalized marker gene expression for endothelial cells subpopulations. **e** Proportion of each endothelial cell subpopulation across conditions (healthy, left; regenerating, middle; embolized, right). **f** Histogram shows comparison of LSECs (periportal, midzonal, and pericentral LSECs) pseudozonation across conditions. LSECs from regenerating and embolized conditions were projected onto a reference healthy zonation trajectory. **g** Distribution of each LSEC subpopulation from healthy, regenerating and embolized condition is shown along the pseudozonation trajectory. Source data are provided as a Source Data file for (**b**, **c**–**g**).

CYP3A4 showed a weaker expression in PVE tissues between the intermediate lobule area and the central vein. While SAA1/2 (portal marker) diminished nearly completely in regenerating liver samples, the expression of HAL (portal marker) was reduced to areas around the portal fields in both PVE tissues. We also quantified cell density along the central-to-portal axis, and found that, while healthy liver tissue presented uniformly distributed cells, lobules of PVE samples showed variation in cell density (Fig. 3g). We observed a cell density increase in the midzone of regenerated lobules, as well as a drastic loss in central cell density in embolized liver samples.

These results show that, while the main transcriptomic and protein signatures associated with hepatocyte zonation are maintained, the post-PVE human liver is characterized by alterations to hepatocyte metabolism and lobule architecture, in particular a reduced density of pericentral hepatocytes in embolized tissues.

### Altered zonation of endothelial cell state after PVE
We next examined zonation in LSECs after PVE (Fig. 4a). Our analysis in healthy fresh and frozen tissues revealed periportal and pericentral populations consistent with previous reports[7,9,12] (Supplementary Fig. 4). Annotation of liver endothelial cell subpopulations in the combined healthy and post-PVE datasets (Fig. 4b) revealed 9 molecularly distinct LSEC subpopulations, as well as clusters of non-LSEC ECs, lymphatic ECs, and ECs in the G2M/S phase of the cell cycles (Fig. 4c, d). Based on marker gene expression, we cataloged the LSEC subpopulations as periportal (*MGP*, *AQP1*, *IGFBP7*, *CLEC14A*),

pericentral (*CLEC1B*, *CLEC4G*, *CLEC4M*, and *FRZB*), midzonal (periportal/pericentral markers), fenestrated (*PLVAP*, *RBP7*), remodeling (*CTGF*, *IGFBP3*, *ANGPT2*), interferon (*CXCL10*, *IFI44L*, *ISG15*, *IFIT3*). We also identified three subpopulations expressing mitochondrial and stress-associated genes, which may be associated with tissue processing or alternatively a previously described relationship with augmented shear stress in endothelial cells during liver regeneration[30].

We observed differential abundance of pericentral LSEC populations, with more cycling cells and high mitochondrial content cells in the regenerating condition, and an enrichment of interferon and *ANGPT2*+ remodeling cells in the embolized condition (Fig. 4e). Pseudozonation trajectory alignment to the healthy reference suggested a moderate shift towards pericentral identity in the regenerating liver, whereas LSECs from embolized samples had a slight inclination towards a periportal expression pattern (Fig. 4f, Supplementary Fig. 11a, b). Regenerating and embolized LSEC, when compared to healthy, both upregulated genes related to developmental, cell adhesion, and migration programs (Supplementary Fig. 11c, d). Zonation of various genes also differed between conditions (Supplementary Fig. 11e, Supplementary Data 6), yet these did not translate into any enriched GO Terms.

Projection of LSEC populations to the healthy pseudozonation reference revealed that remodeling, interferon-high, and stress-associated LSECs primarily map pericentrally, while fenestrated LSECs had a predominantly periportal signature (Fig. 4g). Fenestrated cells had a strong matching signature to scar-associated (SA)

endothelial cells with a strong pro-fibrinogenic signature[13] (Supplementary Fig. 11f), highlighting potential fibrosis appearing near portal vessels in the post-PVE liver.

**Modulation of PVE response by immune and mesenchymal cells**
Previous scRNA-seq studies have shown the large diversity of immune cells in the human liver[9,12,13,15]. Hepatic immune cells are crucial for maintaining liver homeostasis and regulating regeneration[31]. In our dataset, we identified 24 cell clusters across healthy and PVE liver tissues, belonging to myeloid and lymphoid lineages (Fig. 5a–c), featuring well-defined cell types identifiable by their known markers (Supplementary Fig. 12a, b). Some populations often showed an enrichment or depletion in PVE samples, compared to the healthy condition (Supplementary Fig. 12c). We detected an increase in NK cells in regenerating tissue (Fig. 5d, Supplementary Fig. 12c), and an increase in plasmacytoid dendritic cells (pDCs) in embolized tissue (Fig. 5d). In the myeloid compartment, we observed a decrease of Kupffer cell proportions in post-PVE tissues, accompanied by an increase in other Monocyte/Macrophage populations (Fig. 5d), such as those expressing *IGSF21*, a gene with a proinflammatory role[32]; and TREM2+ Monocytes, which have previously been identified as a scar-associated macrophage subtype. These observations are supported by probing these populations in the three conditions for a gene expression signature obtained from an independent liver cirrhosis dataset[13] (Fig. 5e), with both matching the more pro-fibrinogenic TREM2+ signature (SAMac (2)).

We then investigated how different cell populations together orchestrate repair and regrowth of hepatic tissue, by mining changes to cell-cell interactions[33] in post-PVE tissue. Most predicted interactions involved macrophages, endothelial, or mesenchymal cells (Fig. 5f, Supplementary Fig. 13a). These cell populations displayed the largest amount of non-homotypic interactions (Supplementary Figs. 13b and 11c), as well as the largest increases in number of interactions in post-PVE liver, particularly in the regenerating sections (Supplementary Fig. 13b, c). Most cell types showed an increased number of cell-cell interactions in the regenerating tissue, the largest seen in pDCs, dividing ECs, and pericentral LSECs. Curiously, hepatocytes showed a modest reduction of interactions in post-PVE samples compared to healthy.

In order to reveal changes to the global interaction network, genes belonging to ligand-receptor pairs were summarized into a correlation-based co-expression network and projected in 2D using UMAP[34], providing a clear layout for the network of combined post-PVE and healthy tissues and each condition separately (Fig. 5g, Supplementary Fig. 13b). This highlighted a group of characteristically mesenchymal interactions, putting this cell type at the center of liver cell-cell interactions, despite the low number of detected cells. Together with LSEC, mesenchymal cells constituted a major communication axis after PVE. LSEC post-PVE signaling events included angiocrine-related interactions, which were particularly expressed in periportal populations (Supplementary Fig. 14). We also observed immune-related interactions present post-PVE, such as *CCL18* expression by TREM2+ monocytes (Supplementary Fig. 15). This cytokine was predicted to interact with *CCR1* present in other monocytes and T cells (only in embolized), likely eliciting an inflammatory response. Despite the lower number of T cell-associated interactions in post-PVE liver, this may demonstrate a potential role for rare cell populations in liver regeneration.

Most interactions involved Extracellular Matrix (ECM) components—Collagens, Fibronectin (*FN1*), Tenascin (*TNC*), and Vitronectin (*VTN*) were found to be more highly expressed in post-PVE samples (Fig. 5h, Supplementary Fig. 15). Hepatocytes from all three lobular zones were found to be producing *FN1* and *VTN*, and mesenchymal cells expressed *COL12A1*, *COL5A3*, as well as *TNC*. We confirmed by immunohistochemistry that *COL12A1* and *FN1* were more present around periportal vessels, identified by vessel morphology and co-localization with *KRT19* (Fig. 5i). Throughout the imaged slices,

expression of both ECM proteins (normalized by DAPI signal) was higher in regenerating and embolized samples (Fig. 5j). Various types of endothelial cells interacted with these matrix proteins, as well as other receptors related to cell migration and angiogenesis (*EPHA3*, *NOTCH1*, Supplementary Fig. 15), suggesting a role for mesenchymal cells and hepatocytes in improving vascularization of the regenerating liver through ECM modulation.

**Multiplexed immunofluorescence illuminates PVE liver lobules**
We established an experimental and computational pipeline to perform iterative indirect immunofluorescence imaging (4i)[35] on large (average area = 0.25 mm$^2$) sections of healthy, regenerating, and embolized human liver tissues (Fig. 6a, see Methods). Tiled images were acquired for each section in multiple staining and imaging cycles. Image registration across cycles and data quality control resulted in a dataset of over 1 million pixels covering 16 antibody stains. This method covered centimeter to micrometer length scales (Fig. 6b), enabling protein stain and histological feature analysis from individual cells to patterns across liver lobules and vessels (Fig. 6c, Supplementary Fig. 16a). Complemented by observation of tissue structures (e.g., portal triads and central vessels), the antibody panel resolved hepatocyte zonation, sinusoids, stromal elements, immune cell location, bile ducts, and blood vessel areas (Fig. 6d).

Analysis of post-PVE samples provided several protein and morphological features to characterize regenerating and embolized tissue with similar resolution to healthy tissue (Fig. 6d–f, Supplementary Fig. 16a–c). To explore the microenvironment surrounding vessels across conditions, we established a computational pipeline to segment vessel areas and the nuclei within these areas (Fig. 6g, Supplementary Fig. 16d–f, see Methods). Periportal and pericentral markers in hepatocytes and LSEC were used to distinguish portal and central vessels (Supplementary Fig. 16g–i), verified by morphological inspection of various examples (Fig. 6i). Normalized expression per vessel revealed an increase in ACTA2, a marker for mesenchymal cells, in post-PVE conditions (Fig. 6g, bottom right). In each condition, major cell types—hepatocytes, LSECs, immune cells, and vessel stroma cells-were identified through clustering of marker proteins and other nucleus physical features (Fig. 6h, Supplementary Fig. 17a–c, see Methods). A differential distribution of cell types around vessels was observed (Fig. 6i, Supplementary Fig. 17d–f), with a significant increase of stromal cells in PVE samples (Fig. 6j), particularly around portal vessels. This coincided with an increase in the number of neighboring cells in regenerating portal vessels for all cell types except hepatocytes (Supplementary Fig. 17g). Furthermore, immune cells in regenerating tissue, unlike in healthy or embolized, had a higher proportion of vessel stroma cells as immediate neighbors (Supplementary Fig. 17h). These results highlight PVE-induced liver lobule reorganization, and support observations from scRNA-seq data that immune-endothelial-mesenchymal interactions underlie architectural changes in the regenerating tissue.

## Discussion
Animal models have provided information on the cellular sources and molecular pathways operating during liver regeneration[20], including through the use of scRNA-seq[21,36–38]. Nonetheless, there is a lack of descriptive knowledge from primary human tissues that have undergone regeneration[19]. These gaps are mainly due to technical, logistic, and ethical challenges associated with sample collection, processing, storage, and time-dependent experimentation.

We established experimental protocols for surveying cell states in healthy fresh and frozen human liver samples, similar to two published methods[16,17], showing the power and limitations of each approach. Our optimized protocol will allow access to frozen liver tissue banks and facilitate a more flexible investigation into regenerative processes, hepatic diseases, and malignancies. While

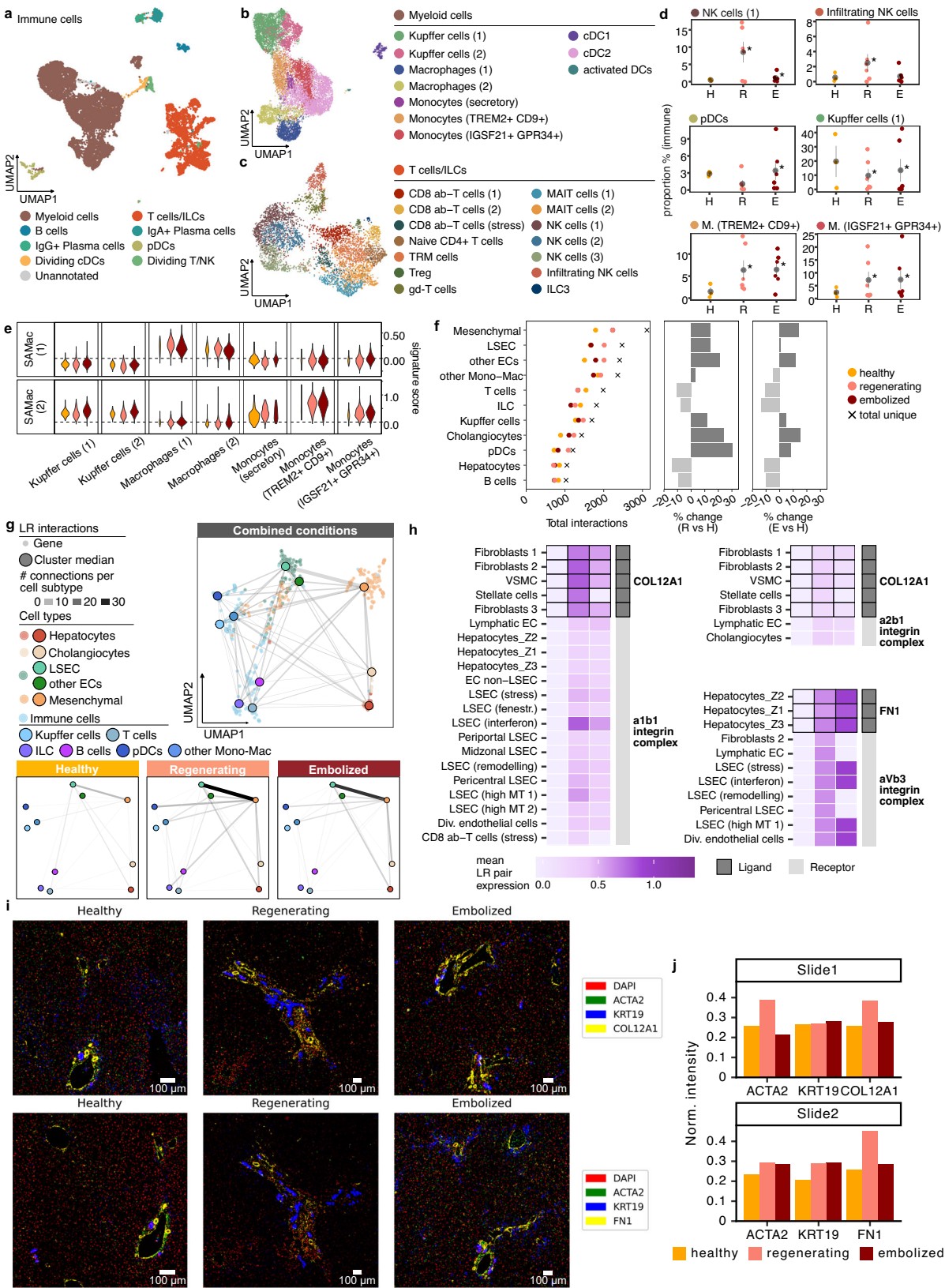

this approach resulted in a lower number of detected genes per cell, it was capable of capturing more nuclei from cell types that were lowly represented in data generated from fresh samples. Fresh and frozen tissue data complement each other and suggest that a global cell type census should not rely on a single isolation method.

We produced a human liver regeneration transcriptomic atlas with over 100,000 cells of healthy and post-PVE hepatic tissues, resulting in a detailed census of cell states in healthy, regenerating, and embolized liver tissue. This encompasses different snapshots of the regenerative process from multiple post-PVE sample collection timings. Despite not covering the earlier days post-PVE, we demonstrate

**Fig. 5 | Intercellular signaling map reveals fibroblast and immune-cell coordination of response to PVE.** UMAP plots for all immune cells (**a**), and the myeloid (**b**) and lymphoid (**c**) subsets, colored by the respective annotations. **d** Changes in proportions for select immune cell populations. Gray point shows mean and standard error intervals. * denotes a significant difference compared to healthy (two-tailed binomial test, *p* < 0.05). H healthy, R regenerating, E embolized. **e** Scar-associated macrophage (SAMac) signature scores in myeloid cell populations, in each condition. **f** Number of interactions involving each major cell type, in each condition and in total (left dotplot), along with variation in the number of interactions between healthy and regenerating (middle) or embolized (right). **g** UMAP plot showing all and condition-specific ligand-receptor interaction networks, summarized by cell type. **h** Heatmaps showing mean expression for selected ligand-receptor pairs involving ECM proteins across healthy, regenerating, and embolized conditions. Black outline denotes the cell type uniquely expressing the ligand. Gray sidebar distinguishes expression of ligand and receptor. Additional heatmaps are shown in Supplementary Fig. 15. **i** Regions of Interest from immunohistochemistry slides for two selected ECM proteins. **j** Total fluorescence intensity of the three proteins assessed in each immunohistochemistry slide, normalized by total DAPI intensity in the tissue area. Source data are provided as a Source Data file for (**a–h**).

that signatures related to angiogenesis and other developmental processes are still present in cholangiocytes and endothelial cells, which are of crucial importance to re-establishing liver lobule structure and function. Notably, regenerating tissue hepatocytes showed less pronounced changes than cholangiocytes, endothelial and immune cells, suggesting that hepatocytes may shift more rapidly towards a homeostatic state. Identification of type I interferon response and cytokine signaling pathways in the regenerating liver further suggested a termination of hepatocyte proliferation[39], and thus a late regeneration stage of our samples.

PVE resulted in disruption of hepatocyte zonation, with alterations of zonation profiles for lipid metabolism and energy production pathways. We also predicted an increase in periportal-like hepatocytes in embolized lobules, with only partial recovery in regenerating tissues. These gene expression results matched our observation of more densely packed periportal cells in these tissue sections. This further underscores the importance of the periportal zone and hepatocyte energy metabolism as points of interest for future studies and therapies covering liver damage recovery.

LSEC populations retained zonation profiles in post-PVE tissues. However, we observed differences between conditions in cell state proportions, as well as increased expression of genes involved in stress response, EC remodeling, interferon signaling, and cell cycle regulation, and expansion of endothelial cells in the periportal zone post-PVE. These processes are reminiscent of previous work from mice, in which inductive angiocrine signals from sinusoidal endothelium stimulate hepatocyte proliferation, and the secreted ligand Angiopoietin 2 orchestrating phases of LSEC and hepatocyte proliferation[40,41]. Our data show an increase in angiocrine signaling by periportal LSEC, further underscoring the importance of periportal region remodeling in the regenerating liver.

We uncovered diverse myeloid and lymphoid immune cell subtypes. In the regenerating liver, we saw an increase of infiltrating NK cells, which are associated with negative regulation of liver regeneration[42]. Beyond known roles for myeloid populations in liver regeneration[39,43], we predicted that TREM2+ scar-associated monocytes[13] manage immune cell recruitment to regenerating tissue via *CCL18*, a T cell-attracting chemokine expressed by periportal mononuclear cells[44]. We also observed several ECM-related interactions predicted to be orchestrated by mesenchymal cells (fibroblasts, VSMC, and HSCs), with LSEC and other endothelial cells expressing receptors that recognize these proteins. COL12A1 and FN1, two tested ECM proteins, were expressed around vessels, demonstrating that perivascular mesenchymal cells actively shape liver cell interactions.

PVE elicits responses by the liver across scales, and thus an assessment of this response should capture biological features across tissue, cell, and subcellular levels. We established a multiplexed immunohistochemistry technique (4i) on large tissue sections to obtain a dataset with hundreds of thousands of cells across hundreds of millions of pixels and used the data to explore the vessel area microenvironment. Importantly, the 4i experiment, along with the increased detection of KRT19, expression of ECM proteins in post-PVE portal vessels, and scRNA-seq-based inferences on hepatocyte and LSEC zonation and cell-cell communication, have underscored

periportal vessel region as the main site of liver lobule reorganization and regeneration. While more specific cell types could not be identified, future experiments can focus on unraveling the full intercellular interaction network in spatially defined niches.

Understanding hepatic growth, regeneration, and degeneration is a topic of growing importance in the medical and bioengineering fields[45,46]. Our work delivers a human PVE atlas which serves as a blueprint for future studies on the mechanisms of human liver regeneration, and to formulate hypotheses to steer regenerative and developmental states using liver organoid model systems[45,47,48]. These data revealed that diverse cell states and interactions were induced in the hypertrophic and atrophic conditions after PVE and that therapeutic modulation of cell-cell interactions may be harnessed to enhance desired patient outcomes.

## Methods

### Experimental model

**Human liver tissue samples.** Human adult liver tissue samples were obtained from macroscopically "healthy" tissue that remained from resected human liver of patients with primary or secondary liver tumors or benign liver diseases with or without pretreatment with PVE. Participants of this study gave their informed consent that their tissue samples and patient data (sex, age, diagnosis) can be used after pseudonymization for research purposes and publication, according to the ethical guidelines of Leipzig University Hospital (006/17-ek, 21 March 2017, revised and renewed 12 February 2019). Acquired tissue from portal vein embolized livers included samples of regenerating (R) and embolized (E) tissue whereas tissue samples from benign liver diseases were defined as quiescent healthy controls (H) (Supplementary Data 1). Donors have received no compensation for participating in this study.

### Experiments

**Isolation of human liver cells from fresh tissue.** Isolation of the primary human hepatocytes (PHH) and non-parenchymal cells (NPC) from the liver tissue was performed as described previously[49]. Briefly, PHH and NPC were isolated from the same tissue sample simultaneously by a two-step EDTA/collagenase perfusion technique and then purified by Percoll density gradient centrifugation. To obtain different cell types from NPC fraction, this cell suspension underwent two different centrifugation steps: $300\,g$ for 5 min to get liver endothelial cells, mesenchymal cells and Kupffer cells; $650\,g$ for 7 min to get the majority of Kupffer cells. Finally, 3 cell suspensions, PHH, NPC300 and NPC650, were used to prepare a single-cell RNA-seq experiment. In case of samples with medical conditions, the regenerating, and embolized tissues, the isolation procedure was made in the same manner.

**Single-cell suspension preparation and single-cell RNA-seq experiment.** Single-cell RNA-seq experiments were performed using a 10X Genomics platform. Before loading on a microfluidic chip, suspensions of PHH and both NPC fractions were washed and filtered at least twice in ice-cold 1X HBSS without $Ca^{2+}$ and $Mg^{2+}$ (HBSS w/o, Sigma) to remove tissue and cellular debris and to get individual cells

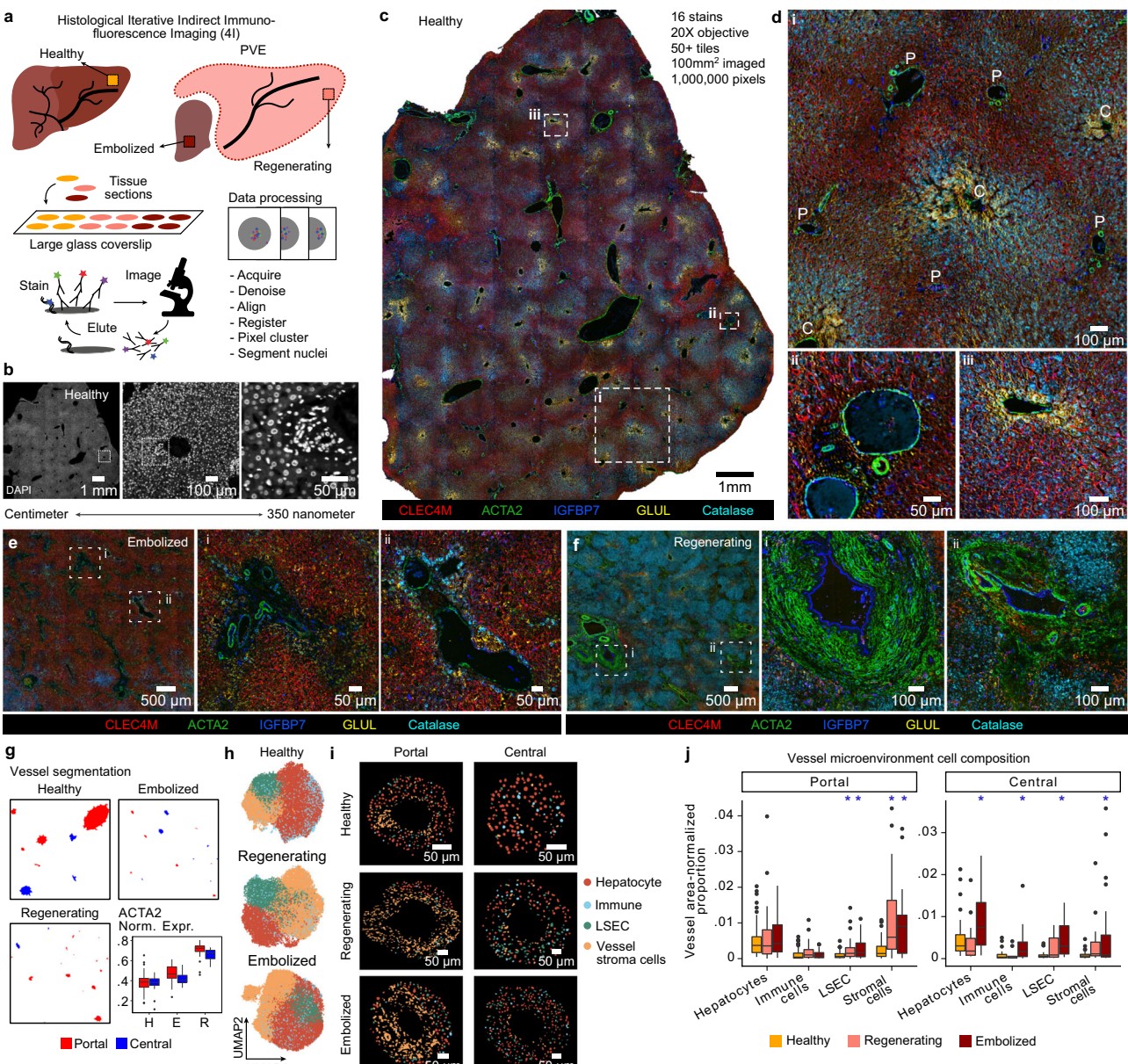

**Fig. 6 | Multiplexed immunohistochemistry reveals peri-vessel microenvironment alterations in post-PVE patient tissue sections. a** Schematic illustrating the tissue-level 4i protocol. **b** DAPI staining of healthy liver tissue, illustrating the image resolution across length scales. Multiple staining overlays (expression of CLEC4M, ACTA2, IGFBP7, GLUL, Catalase representing pericentral LSECs, mesenchymal cells, periportal LSECs, hepatocytes, and peroxisomes, respectively) in a large healthy liver tissue section (**c**) and selected highlighted regions (**d**), where (i) shows an annotated liver lobule (P portal, C central), and the bottom panels a close-up of a portal (ii) and central (iii) vessel. Multiple staining overlays in large and highlighted sections of embolized (**e**) and regenerating (**f**) liver samples. **g** Portal and central vessel detection results from liver 4i data (Methods, Supplementary Fig. 16). Boxplots (bottom right) show normalized expression of ACTA2 per vessel and

condition. Boxplot shows the median (center line), 25th, and 75th percentile (lower and upper boundary), and the whiskers indicate the 1.5 × inter-quartile range, with outliers shown as individual data points. **h** UMAP plots based on 4i data and derived physical measurements for nuclei segmented in each condition, colored by cell type. **i** Representative portal and central vessels from each condition, with segmented nuclei colored by cell type. **j** Variation in cell type proportions shown as normalized proportion of nuclei per vessel area in portal and central vessel microenvironments across conditions (* indicates two-sided t-test fdr-adjusted $p < 0.05$ between regenerating/embolized and healthy; boxplot shows the median (center line), 25th, and 75th percentile (lower and upper boundary), and the whiskers indicate the 1.5 × inter-quartile range, with outliers shown as individual data points). Source data are provided as a Source Data file for (**h**1–3 and **j**).

in media that is compatible with the downstream experimental steps. Wide-bore pipette tips were used working with PHH to avoid premature cell lysis, while p1,000 and p200 pipettes were used gently resuspending NPC cell pellets.

Preparation steps of the single-cell suspension were made on ice and every time washed samples were spun down using a 4 °C cooled centrifuge: PHH at $50\,g$ for 5 min, NPC300 at $300\,g$ for 5 min, and NPC650 at $650\,g$ for 7 min. Finally, to generate single-cell suspension,

PHH suspension was filtered through 40 and 30 and both NPC fractions through 30 μm diameter cell strainers. Cell viability and concentration were assessed using a cell analyzer (Muse™ Cell Analyzer, Luminex Corporation). NPC300 and NPC650 were pooled 1:1 and loaded on a 1 channel and PHH on a 2 channel of the Single Cell A and B chips targeting 6000–8000 cells per each sample. All steps of the single-cell suspension preparation for the regenerating and embolized tissue samples were executed following the healthy tissue protocol.

The next steps were conducted as described in the Chromium Single Cell 3' Reagent v2 and v3 Kits. In brief, after generation of the droplets with the single cells and barcoded beads, cDNA synthesis was performed. Next, droplets were broken, cDNA was amplified and libraries were constructed with different Chromium i7 Sample indexes in order to record sample assignment during computational analysis. Finally, single-cell libraries were run paired-end (28 bp, 8 bp, 100 bp) on an Illumina HiSeq2500 platform on 2 lanes. Experimental summary metrics can be found in Supplementary Data 8.

**Frozen human liver tissue dissociation into single-nucleus suspension and flow cytometry sorting.** Human frozen liver tissue samples were dissociated into single-nuclei combining liquid homogenization cell lysis with Dounce homogenizer and detergent-based lysis methods. All steps of the nuclei isolation were performed on ice with precooled solutions and using 4 °C mode centrifugation. The dissociation protocol that was previously used on brain tissue[50], was optimized here to maximize the nuclei isolation for the liver tissue. The protocol after the optimization included the following steps: first, thawing tissue sample was cut into smaller pieces, minced and transferred into a glass dounce homogenizer. 30 strokes of pestle A were used to homogenize the tissue in 0.3 M Sucrose (Sigma) solution including 0.002 M EDTA (Thermo Scientific), 1% BSA (Serva), and 1% Tergitol solution (Sigma). After 5 min of incubation, next 30 strokes of pestle B were applied to finalize the disruption process and deliberate nuclei from the cells into suspension. Homogenized solution was centrifuged at 600 g for 5 min and the nuclei pellet was washed twice in PBS solution (0.002 M EDTA, 1% BSA, 0.2 U/ul RNase Inhibitor (Thermo Scientific)). Finally, to remove any aggregate and debris the nucleus suspension was filtered through a 30 μm diameter strainer and resuspended in PBS solution (1% BSA, 0.2 U/ul RNase Inhibitor). Further, to enrich for individual nuclei, the suspension was sorted by applying a 4-way purity mode based on the selected DAPI positive nuclei population (1:1000, BD Pharmingen) using forward and side scatter gating strategy (FACS). These nuclei were sorted in bulk and kept on ice for >30 min. To ensure that the sorted nuclei were intact, they were stained with DAPI (1:500) to inspect under the fluorescence microscope and finally counted using a hemocytometer before a single-nucleus RNA-seq experiment.

**Single-nucleus RNA-seq experiment.** Sorted single-nucleus suspensions were loaded on a Single Cell B chip to generate single-nucleus gel beads in emulsion on a 10X Chromium controller. Single-nucleus RNA-seq libraries were prepared following the protocol of the Single Cell 3' Reagent Kit v3 and sequenced paired-end (28 bp, 8 bp, 100 bp) using an Illumina HiSeq2500 platform.

**Cross-sectional Imaging.** For liver segmentation and volume measurement 3D-datasets of both computed tomography (CT) and magnetic resonance imaging (MRI) were used.

CT was acquired as a 128-slice multidetector helical intravenously contrast-enhanced (10–120 mL iomeprol; Imeron 400, Bracco, Milan, Italy) scanner (Ingenuity, Philips, Best, The Netherlands). Contrast phase scanning was adjusted to the necessity of visualization of all liver architecture relevant for segmentation (i.e., liver veins, portal vein branches (scan delay of 70–90 s after injection). Primarily axial reconstruction of images in 1–2.5 mm slice thickness (increment, 1)

MRI (1.5T, Magnetom Aera, Siemens, Erlangen, Germany) was acquired with the use of 0, 1 mL/kg body weight gadoxetic acid (Primovist/Eovist, Bayer, Leverkusen, Germany), in the early dynamic phases and negative contrast of blood vessels in the hepatobiliary phase (15–25 min. delay after injection) and a fat saturated breath hold T1-weighted interpolated 3D-sequence (VIBE).

All image data sets were digitally archived (DICOM format), pseudoanonymized, and exported to a dedicated post-processing

workstation (Lenovo ThinkStation, Lenovo, Beijing, China) inside the institutional network.

Volume segmentation was done semi-automatically by outlining the liver surface excluding large hilar vessels and interceptions with a dedicated post-processing tool (3D Slicer, open-source software). Virtual resection along anatomic landmarks were performed to assess the future liver remnant before and after the PVE regarding the outcome parameters volume, vessel architecture, and tumor progression. 3D-Visualization of the 3D model was performed using the Blender software (Blender Foundation, Amsterdam, Netherlands).

**H/E staining and Immunohistochemistry.** For investigation of tissue sections human liver tissue samples ($n = 3$ per condition, Supplementary Fig. 7) were fixed with paraformaldehyde (PFA, Carl Roth, Karlsruhe, Germany), embedded in paraffin, sectioned into 3.5 μm thick slices using a microtome (MicromHM430, Thermo Fisher Scientific, Waltham, MA, USA), and mounted on slides.

For Hematoxylin and Eosin (H&E) staining the tissue sections were rehydrated. Then the tissue sections were incubated in Hämalaun (Merck, Darmstadt, Germany) for nuclei staining for 5 min at RT and washed under rinsing water for 10 min. Afterwards the tissue sections were stained with Eosin G-solution (Merck, Darmstadt, Germany) for 3 min. Finally, the tissue sections were dehydrated and the slides were embedded in the non-aqueous mounting medium Entellan (Merck, Darmstadt, Germany).

The immunostaining of tissue sections and their microscopic evaluation was performed as described previously[51] with the following modifications. Briefly, tissue sections were rehydrated and epitopes were retrieved. Then, tissue slices were blocked for endogenous peroxidase activities and for unspecific binding. For detection of GLUL, HAL, SAA12, IGFBP2, CYP3A4, CK19, CD31 specific primary antibodies (all Abcam, United Kingdom, Cambridge) were used (Supplementary Data 9). Antibodies were diluted in TBS (Sigma, Munich, Germany) with 1% BSA (Sigma, Munich, Germany) and 0.03% TritonX-100 (Sigma, Munich, Germany).

The antibodies against the targets were visualized using Peroxidase-conjugated secondary antibodies (Supplementary Data 9). Secondary antibodies were diluted in TBST (TBS + 0.5% Tween20 (Sigma, Munich, Germany)) supplemented with 1% BSA. Detection was performed by using 3,30-Diaminobenzidine (DAB, Sigma, Munich, Germany).

The EnVision+Dual Link System-HRP (Dako, Glostup, Denmark) was used according to manufacturer instructions when the targets showed low expression in the tissue sections. All reactions were stopped and cell nuclei were stained with hematoxylin. Finally, the slices were dehydrated and embedded using Entellan (Merck, Darmstadt, Germany).

For immunofluorescence (IF) staining the above described procedure was repeated with the following modifications. Primary antibodies were diluted in TBS with 1% BSA and 0.1% Tween20. Further, the staining was performed without blocking for peroxidases. For visualization the secondary antibodies ALEXA Fluor 647 donkey anti rabbit and ALEXA Fluor 488 donkey anti mouse (both Abcam, United Kingdom, Cambridge) were used and diluted as described above. Cell nuclei were stained with Hoechst. Finally, the slides were embedded in Mowiol-488 (Carl Roth, Karlsruhe, Germany). Negative controls for DAB and IF stainings were made from all donors and treated in the same way but without usage of a primary antibody.

**Zonation imaging analysis.** Whole slide images of IF stainings were captured in fluorescence mode using a Slide Scanner (AxioScan Z1, Carl Zeiss, Oberkochen, Germany) with a 20/0.8 M27 Plan-Apochromat objective with channel 1 (Target) light source 630 nm, light source intensity 50% with extinction wavelength 631 nm and emission wavelength 647 nm and with channel 2 (Hoechst) fluorescence light source

385 nm, light source intensity 14,53% with extinction wavelength 353 nm and emission wavelength 465 nm with an Axiocam 506 m as imaging device. Resulting images were stored as raw data in the Carl Zeiss proprietary image pyramid format (CZI) with an object-related nominal pixel size of 0.227 μm × 0.227 μm.

Whole slide images of DAB and H/E stainings were captured in transmitted light mode using a Slide Scanner (Pannoramic Scan 2, 3DHISTECH, Budapest, Hungary) with a 20/1.6 Plan-Apochromat objective and stored as raw data in the MIRAX Virtual Slide Format (MRXS) with an object-related nominal pixel size of 0.243 μm × 0.243 μm.

Tissue sections were visually investigated for lobular architecture ($n$ = 3 per condition, analog to Supplementary Fig. 10). For that we used sequenced sections from immunofluorescence stainings for GLUL and Hoechst (pericentral hepatocytes and cell nuclei) and HAL and Hoechst (periportal hepatocytes and cell nuclei), complemented by observation of tissue structures (e.g., portal triads and central vessels) to identify the lobular vessel architecture and lobular borders. For the further imaging analysis, the Hoechst channel from the GLUL staining was used. Here, feature regions containing identifiable liver lobules were stored from Zeiss ZEN in 10,000 × 10,000 Pixel Tiles as Portable Network Graphic (PNG). Clearly identifiable lobules were donor and condition dependent resulting in $n$ = 15 lobules for healthy, $n$ = 12 lobules for regenerated and $n$ = 10 lobules for embolized tissue, respectively. Fiji (ImageJ) was used to preprocess the selected feature regions of the tissue. In a first step the coordinates of the lobule boundaries and the location of the central vein were set by hand and stored as "Comma Separated Values" (CSV) files for later imaging processing. Between each coordinate of the lobule boundaries and its corresponding central vein a portal-central axis results. The number of axes was at least six but finally dependent on lobule shape complexity and was tried to keep as close to six as achievable.Then a gaussian filtered version (sigma = 20) of the image was subtracted from the original to correct staining artifacts and normalize overexposed regions and the background. After that the image was stored as "Tagged Image File Format" (TIFF). The density investigation started with an approximate estimate of the nuclei centers and measurement of their spatial locations. For that an Otsu threshold was used to filter the detected nuclei and reduce the amount of nonspecific signals resulting in a binarized image. The feature region of the nuclei in the binarized image were processed with morphological filters to remove small white noise and fill holes. For that we use morphological opening and closing. Regions near to the center of a feature region are defined as foreground and regions far away to a feature region are defined as background. The distance transformation provides the pixels known as certain of a region that belongs to a nucleus. With a weighted threshold of the maximum distance transformation value we received sure foreground regions of an identified nucleus indicated as a marker. A connected component analysis labels these regions with any positive integers to separate these from the background. Processing these markers with a marker-based watershed method allows to allocate the unclassified regions of the image belonging to nuclei regions or not. The resulting region property provides sure coordinates of spatial locations of nuclei.

To get the spatial dependent location density we use the most popular spatial analysis technique called kernel density estimation (KDE). KDE is a statistical method and a non-parametric way to estimate the probability density function of a random feature and correlates it to features in its neighborhood. By correlating each feature with a kernel function and summing up all the weighted overlapping regions of the kernel we get the probability density as a level of spatial distribution of nuclei density. The kernel is chosen as a multivariate gauss function with a fixed bandwidth of 0.2 for all images to ensure comparability of lobules from different sections among each other. The bandwith was empirically determined by analyzing 3 of

the control samples by using the scott algorithm for bandwidth estimation.

The resulting spatial density distribution of nuclei was plotted as a heatmap. In a next step we transferred the lobule coordinates into the heatmap and extracted the density values along the earlier defined portal-central axis of a lobule ($n$ = 121 axes for healthy, $n$ = 84 axes for regenerated tissue and $n$ = 76 axes for embolized tissue). The portal-central density distribution of a condition was plotted as the mean density values along the portal-central axes including standard deviation.

**Iterative indirect immunofluorescence imaging (4i) experiment.** 4i experiments were performed on 3μm thick formalin fixed paraffin embedded (FFPE) sections that were arranged on a large (110 × 75 mm) H1.5 glass plate (Schott Nexterion 1535661). The glass surface was silanized using 3-aminopropyltriethoxysilane (aptes). The tissue was non-covalently bound to the functionalized surface with 10% gluteraldehyde before deparaffinization with NeoClear and reductive Ethanol baths. A second fixation with 4% PFA was performed, followed by blocking of aldehydes with 50 mM NH4Cl. Heat induced epitope retrieval in Citrate Buffer (10 mM Citric Acid, 0.05% Tween20, pH 6.0) heated over 20 to 90 °C in a histological microwave (Milestone RHS1).

The glass plate with tissue sections was then mounted on a custom printed PLA superstructure (Prusa i3MK3S+ printer) to create in essence a single well SBS-format plate with an imageable area of 100 × 65 mm.

On this assembly iterative indirect immunofluorescence imaging (4i) was performed according to the method by G. Gut[35] with volumes adapted to the well size. The plate was handled shielded from direct light until the imaging buffer was added. All incubations were done on an orbital shaker (160 rpm, 20 mm orbital diameter). In total, our assay comprised 36 antibodies and a nuclear stain.

The iterative immunofluorescence was done on a Nikon Ti2 automated microscope sided by a Crest X-light V3 spinning disc and a Lumencor Celesta light engine with a Nikon 20x water immersion NA 0.95 (MRD77200) to cover length scales from the millimeter to the micrometer scale (pixel size 325 nm) (Fig. 6a). The single sections were bounded in tiled regions to facilitate the later stitching of large images.

Image pre-processing (maximum intensity projection, camera baseline subtraction, shading correction, and stitching) was carried out with ImageJ batch processing macros.

**4i data processing.** Image data processing broadly followed the steps in ref. 52. Briefly, we first produced a rough mask to align each image. Then, we aligned full tissue stitched images by relying on the DAPI channel present in every cycle, using Elastix implementation in the SimpleITK python package[53], with each image aligned to the preceding elution cycle. This was followed by image cropping and denoising using the scikit-image package[54], followed by the production of a refined full tissue mask. Lastly, we performed background subtraction, using for images in each cycle both the preceding and succeeding elution cycles as a reference, weighted by the distance of the cycle to be corrected to each elution cycle.

**4i data - pixel clustering and blood vessel detection.** Images were converted into pixel-by-antibody matrices, and each image was additionally downsampled to 0.1% of the total pixels to facilitate handling and analysis. Following normalization, log-scaling, and standardization, the downsampled matrices were then used to obtain clusters of pixels using scanpy[55]. Leiden[56] clustering was used with a resolution of 0.3, with 50 neighbors calculated on the top 10 principal components.

This clustering was then used to subset 80.000 pixels, proportionally weighted for each label, from each image. These pixels were then normalized, log-scaled and standardized. Data from the three samples was integrated using Harmony[57], and the top 10

components were used for finding neighbors, followed by Leiden clustering with resolution 0.5. This was then propagated to the remaining pixels by assigning them with the label from the closest labeled pixel using euclidean distance.

This clustering resulted in a uniform labeling of pixels across the three images. Labels 8, 9, 10, and 11 were selected as being potentially associated with blood vessels. For all samples, the pixels representing the union of these labels were refined using scikit-image to remove small objects and smooth the vessel mask. In addition, for the regenerating sample clusters, a blob detection algorithm was applied to resolve the centroids of vessels in very close proximity and segment them accordingly.

For each individual vessel, the mean signal for each antibody was calculated, as well as the eccentricity of the vessel's shape. These vessels-by-features tables were used in scanpy to cluster the vessels in each sample to remove regions incorrectly identified as portal or central vessels, such as some tissue margins.

Lastly, to home in on the vessel proper, we used the nuclei segmentation mask (see next section) and scikit-image to find areas without nuclei. These regions were intersected with the vessel areas identified previously to obtain the focused vessel region.

**4i data—nuclei-centered analysis.** Nuclei masks for each sample were produced using cellpose[58] based on the DAPI channel from the first elution cycle. These masks were used to define each nucleus' position and the pixels assigned to them. Pixels corresponding to the cytoplasm were obtained by capturing a region of up to 5 pixels around each nuclei. In order to focus the analysis on the vessel microenvironment, only nuclei within a 300 pixel radius were considered for analysis.

The nuclear and cytoplasmic regions were used to collect various metrics. In addition to the mean antibody signal, normalized to a 0 to 1 interval, in each region, we also collected information on area and nuclear eccentricity, as well as the number of neighbors each nuclei had within a radius of 200 pixels. For the remaining analysis, the signal from the following antibodies was kept: CD45 (Abcam, ab10558), CLEC4M (Origene, CF810055), TROP2 (Invitrogen, PA5-47074), ACTA2 (Sigma, A2547), IGFBP7 (Invitrogen, PA5-47123), GLUL (Abcam, ab125724), CRP (Bethyl Laboratories, IHC-00613), LaminB1 (Abcam, ab76024), Catalase (Abcam, ab76024), and e-Cadherin (Abcam, ab11512).

Data for each sample was independently analyzed with scanpy[55]. All data was first standardized, followed by regressing out of the total antibody signal in the nuclei and cytoplasmic area, as well as the x and y coordinates of each nucleus in the tissue, to mitigate potential heterogeneities caused by uneven signal distribution across the large tissue area. A neighbors graph was calculated on all features, and this was used to generate a UMAP and perform Leiden clustering with resolution 0.8.

Nuclei were annotated into four major cell types based on a combination of these clusters and thresholds for various markers. This approach was chosen since the rarer immune cells, which were directly observable in the tissue from their CD45 signal, tended to group with other cell types, likely due to capturing neighboring signal from other co-locating cell types. For each cell type, a threshold of 0.5 standardized value was evaluated for different variables: hepatocytes were defined as having high Catalase, GLUL or CRP, high area, low eccentricity, and low CD45 signal; LSECs were defined as having high CLEC4M, CLEC14A, ISG15 or eccentricity, and low ACTA2 and TROP2; vessel stroma cells were defined as having high ACTA2, TROP2 or eccentricity, and low CD45; and immune cells were defined as having high CD45, low eccentricity and low TROP2. Since this still resulted in some ties, a label of immune cell was given precedence over hepatocytes, which in turn was given precedence over vessel stromal cells and LSEC. Lastly, apart from immune cells, the other labels were applied to each Leiden cluster based on a simple majority rule.

To annotate blood vessels as portal or central, we relied on the antibody signal of CRP (portal marker) or GLUL (central marker) in hepatocytes, and IGFBP7 (portal marker) and CLEC4M (central marker) in LSEC and stromal cells. For each nucleus, it was determined which marker (portal or central) was ranked higher in expression, and the nucleus was thus assigned that label. Then, two percentages were calculated for each vessel and each pair of portal/central markers, to determine the proportions of hepatocytes and LSEC+stromal cells that were portal or central. Lastly, vessels were classified as portal if the sum of both portal percentages was higher than that of the central percentages, and vice-versa.

For the cell type composition comparisons, a t-test was used with FDR correction, considering a significance threshold of 0.05. To avoid including nuclei not belonging to the microenvironment surrounding the vessel field proper (due to some identified vessels not originating from a perpendicular cut), only nuclei from vessels within 2 standard deviations of the mean log2 area were considered.

### Single-cell RNA-seq computational analysis

**Data processing after sequencing (Cell Ranger pipeline).** We used the Cell Ranger software (https://support.10xgenomics.com/single-cell-gene-expression/software/overview/welcome) to process the sequenced RNA libraries and generate gene expression count matrices for the analysis. We first transformed Illumina intensities, raw base call (BCL) files into reads using cellranger mkfastq. Next, we ran cell ranger count to align the reads to the human reference genome (GRCh38) using RNA-seq aligner STAR with default parameters (Supplementary Data 10). Uniquely mapped reads were based on barcodes and unique molecular identifiers (UMIs) assigned to cells and genes (ENSEMBL release 84) respectively. Read counts for a given gene and cell that are represented by a Chromium cellular barcode and UMI were used as an input for the subsequent expression analyses.

**Single-cell data filtering and normalization.** Prior to any processing, scrublet[59] was used to assign a doublet score to all cells in each fresh tissue dataset. We used the R package Seurat (version 3.0)[60] to process gene expression count matrices. We first applied SCTransform to normalize molecular counts, scale, and identify variable genes[25] within each dataset separately. Cells in each sample were then finely clustered (Louvain algorithm, resolution = 10) and the average doublet score was calculated to identify small groups of similar doublets. Quality control filtering was done by applying the thresholds outlined in Supplementary Data 11.

**Integration of single-cell data.** Samples were analyzed in two groups: healthy only and all conditions (healthy, regenerating, and embolized). Both groups were integrated using CSS[24]. For the healthy data, integration was done using all common genes and the first 30 principal components. For integration of all samples, the top 3,000 variable genes from the healthy, regenerating and embolized samples, as well as the top 100 marker genes from each cell type identified in the healthy dataset were selected. These were used to do a PCA on the full data, of which the top 50 principal components were used. The genes considered allowed a coverage of the biological variability in all conditions and present populations, despite the unbalanced representation of cell types in each sequenced fraction (Hepatocytes and Non-parenchymal cells).

**Dimensionality reduction, clustering, and annotation of single-cell data.** Projection with UMAP[34] and clustering, both for the healthy and for all combined datasets, was performed using all dimensions obtained from CSS.

For the healthy data, clusters were obtained using Louvain clustering with 0.9 resolution, and markers were detected for these populations using Seurat's FindAllMarkers function (pseudocount.use = 0.1,

logfc.threshold = 0.2, adjusted $p < =0.05$). Some clusters (9, 12, 19) were further individually subclustered to identify specific endothelial, T cell, and pDC/B cell populations, respectively. Annotation was done based on the general and subclustering identified markers, which resulted in some smaller clusters being merged under the same label. Cells were also grouped into five "major cell types" (Fig. 1b), and their marker genes were also calculated using Seurat's FindAllMarkers function (pseudocount.use = 0.1, logfc.threshold=0.2, adjusted $p < =0.05$).

Clustering of combined datasets used the Louvain algorithm with 1.1 resolution. Markers were detected for the identified clusters using Seurat's FindAllMarkers function (pseudocount.use = 0.1, logfc.threshold = 0.2, adjusted $p < =0.05$). Clusters 5, and 19, as well as clusters 6, 22, 26, were subclustered to identify more specific types of endothelial cells, macrophages, and T cells, respectively. All clusters and subclusters were annotated using their top marker genes, and resulted in some clusters being merged into the same cell population.

**Identification of LSEC subtypes in the combined single-cell data.** LSECs subsets were identified by reclustering previously annotated ECs in the combined dataset, followed by data renormalization. These cells were then filtered for non-endothelial and doublet populations based on previously reported LSEC marker genes[6,9] and cells expressing genes from other cell types, respectively. Marker genes for each remaining cluster after filtering were determined using Seurat's FindAllMarkers function (pseudocount.use = x, logfc.threshold = x, adjusted $p < =0.05$). This methodology allowed for the identification of bona fide LSECs (periportal, midzonal, and pericentral), as well as LSEC populations with unique expression profiles, cycling endothelial cells, lymphatic ECs and other non-LSEC ECs likely originating from the portal or central veins (Fig. 3).

**Zonation signature creation in healthy, regenerating, and embolized tissue hepatocytes.** Information on expression of previously established human and mouse zonation marker genes[3,12] was used to identify portal- and central-zone hepatocytes within healthy and post-PVE samples. For each of the three conditions we generated a combined zonation expression signature based on portal and central expression markers. For each gene in both gene sets we calculated the z-normalized expression value across all cells. We then transformed the resulting expression values into the range of between 0 and 1 by subtracting the min expression and dividing by the maximum expression per gene across cells. For each zone-specific gene set we calculated the sum of the normalized gene expression values in a given cell. Per cell we generated combined expression signatures by adding the negative portal signatures to the central signatures. Based on these scores clusters in the UMAP were defined to be showing portal or central-specific expression signatures.

For this analysis healthy, regenerating, and embolized tissue hepatocytes were subset individually from the UMAP of combined datasets based on previously annotated cell type markers. Within each dataset, cells were renormalized using the SCTransform function in Seurat. Four PCs were then used to project cells in UMAP space.

**Comparing healthy, regenerating, and embolized liver samples.** For each annotated cell type, Seurat's FindMarkers function was used to obtain the DE genes between pairs of conditions (Fig. 2f). A maximum of 10,000 cells was used for each condition. Additionally, in order to account for the sequencing depth, prior to calculating the DE genes for hepatocytes, the seqgendiff package[61] was used to downsample the UMI counts for all conditions, taking the minimum median of UMI counts of the three conditions as reference (Supplementary Fig. 5b). Genes encoded in the Y chromosome were disregarded−male donors were only present in the regenerating and embolized conditions, as well as DE genes between conditions that have been detected as

marker genes for other cell types that likely appeared due to ambient RNA contamination.

**GO Term enrichment analysis.** Enrichment for GO Terms was performed using the function enrichGO from the clusterProfiler[62] package, using the all terms in the org.Hs.eg.db package database, with a q-value cutoff of 0.05. The genes considered for analysis were previously identified as DE with an adjusted $p \leq 0.05$ and a logFC > 0.3. All genes tested for differential expression were used as a background set for the analysis. For plotting (Fig. 2k−n), significant GO Terms for each condition were clustered based on their gene similarities using hierarchical clustering, and then grouped into 6 clusters. The term with the lowest $p$ value per cluster was chosen as a representative to be plotted.

**Identification and comparison of hepatocyte and LSEC zonation across medical conditions.** A steady-state transcriptomic zonation reference was established by identifying a latent axis ordering healthy hepatocytes (Fig. 3) and LSECs (Fig. 4) independently, using DiffusionMaps from the Destiny package[63].

For hepatocytes, contaminating non-hepatocytes were first filtered, followed by renormalization and PCA. The first 4 PCs were used as input for DiffusionMaps, and the ranked DC1 dimension, normalized to values between 0 and 1, was defined as the healthy hepatocyte pseudozonation trajectory. Genes varying along this trajectory were determined by parametric ANOVA on a Generalized Additive Model meant to predict gene expression dependent on the pseudozonation trajectory, modeled as a natural spline with 3 degrees of freedom.

LSEC zonation was determined by applying DiffusionMaps to the first 10 PCs of bona fide LSEC populations−Periportal, Midzonal, and Pericentral LSECs (Fig. 4). This first step identified a few outlying cells. These were removed, and the remaining data was renormalized and projected with PCA. DiffusionMaps was run on the first 10 PCs, and the identified DPT was used as a pseudozonation trajectory, after ranking and normalization to the 0−1 interval. Genes varying along this trajectory were determined similarly to those for hepatocytes.

Comparison of hepatocyte and LSEC zonation in the regenerating and embolized liver to the established healthy reference was done by selecting the top 1000 varying genes in the healthy pseudozonation and used them to train a generalized additive model to predict the pseudozonation variable (Supplementary Figs. 8a and 11a). This model used a beta distribution for error modeling with a logistic link function, which guaranteed that the predicted trajectory would be in the interpretable 0−1 range. In each condition, varying genes were determined as described above.

Genes varying in the hepatocytes were determined as differing between pairs of conditions if their Spearman's rank correlation coefficient was lower than 0.3. The fitted expression of these genes was clustered using Euclidean distance and the ward.D2 method for healthy vs regenerating and healthy vs embolized, to identify groups of genes differing in similar ways (Fig. 3i−l).

LSEC varying genes were compared between conditions using Spearman's rank correlation on the fitted values. A correlation coefficient greater than 0.3 indicated a similar behavior between conditions, whereas values below that were considered a different or opposite behavior between conditions. To illustrate this, Supplementary Fig. 11e shows the top 30 similar genes of healthy vs regenerating and healthy vs embolized (PCC >= 0.3), as well as the top 35 different genes (PCC < 0.3) of the same comparisons; each group was obtained by clustered using Euclidean distance and ward.D2 method.

**Identifying cell-cell communication events in healthy, regenerating, and embolized liver.** Ligand-receptor pairs mediating cell-cell communication events were detected within each condition using CellPhoneDB (version 2.0)[33], based on the annotated cell types from the complete data integration (Fig. 5b). The detected ligands and

receptors were then used to create two types of projections summarizing cell-cell communication in the healthy and post-PVE liver. We projected a graph showing all correlations greater than 0.3 between all ligands and receptors using multidimensional scaling (Fig. 5e), and summarized in the same coordinate space each cell type as the median coordinates of the ligands and receptors that are expressed in it at the highest level. We also projected the mean expression per cell type and condition of all ligands and receptors using UMAP (Fig. 5f), and identified the interactions that are unique to healthy or both PVE conditions.

**Detecting enriched types of variable interactions per condition.** For each interaction, in each condition, we obtained a vector encoding whether an interaction was detected in a given pair of cell types. We used these vectors to calculate the mutual information between healthy and regenerating and healthy and embolized samples for each interaction. The resulting values were then used for Gene Set Enrichment Analysis[64] to determine enriched or depleted types of interactions (Fig. 5g). Interaction types were manually annotated based on literature searches, and can be found in Supplementary Data 7.

### Single-nucleus RNA-seq computational analysis

**Single-nucleus RNA-seq data filtering, normalization and clustering analysis.** We used the R package Seurat (version 3.0)[60] to process gene expression count matrices. We first applied SCTransform to normalize molecular counts, scale and identify variable genes[25]. After manual inspection we applied per sample minimum and maximum thresholds on the number of detected genes in a given nucleus to exclude both nuclei with low RNA content and potential doublets (sn_H1: >150 and <1700 detected genes; sn_H3: >200 and <2000 genes; sn_H4: >70 and <1100; sn_R2: and sn_E2: >150 and <1200 genes). In addition, we excluded nuclei with more than 10% of UMIs aligning to mitochondrial genes. The number of nuclei used in the analysis for each condition is provided in Supplementary Data 12.

**Integration of the single-nuclei datasets using batch effect correction.** Sample-specific preprocessed datasets were merged based on the 3000 most variable genes Pearson residuals.

**Cell type identification analysis in the merged single-nuclei healthy datasets.** UMAP was used to represent the similarity of gene expression profiles between nuclei in 2D. The clustering of the healthy nuclei data revealed 5 major clusters. Clusters were assigned to cell types based on the presence of cell-type marker genes that showed a significantly higher expression in a given cluster. DE was performed using the Wilcoxon Rank Sum test between each cluster and remaining clusters.

**Identification of zonation within hepatocytes and LSECs of fresh and frozen healthy liver tissues.** We used the expression of previously established human and mouse zone-specific marker genes[3,9,12] to identify portal and central hepatocytes (Supplementary Fig. 3b, c) as well as periportal and pericentral LSECs (Supplementary Fig. 4a, b) in both fresh and frozen tissue datasets. Portal and central expression signatures were calculated separately across these marker genes in each of the two cell types and datasets. Z-normalization was done per gene and the portal and central scores represented the sum across normalized portal or central marker gene expressions in a given nucleus or cell. The signature was then shown on UMAP embeddings of each cell type and processing protocol. For the frozen tissue dataset, a UMAP embedding containing previously annotated cell types was used. For the fresh tissue dataset hepatocytes and endothelial cells were projected using the top 4 or 15 principal components, respectively. Portal and central signatures were then used to define portal and central groups of cells. DE genes were identified between these sub-clusters in fresh and frozen hepatocytes and LSECs, respectively, by

using FindMarkers function in Seurat with logFC threshold being at 0 and other default parameters. Fold changes between both zonation-linked DE analyses were significantly correlated for hepatocytes (Spearman's rho = 0.19, $p = 2.7 \times 10^{-21}$) (Supplementary Fig. 3d) and LSEC (Spearman's rho = 0.2, $p = 4.6 \times 10^{-27}$) (Supplementary Fig. 4c), suggesting that the tested sub-clusters between cells and nuclei shared a portal and central pattern of zonation.

For the hepatocytes the following genes *HAMP, CRP, SDS, NAMPT, HAL, ID1* were identified as being higher expressed and were shared between both datasets in portal sub-clusters (logFC > 0.4 for fresh and logFC > 1 for frozen) (Supplementary Fig. 3d, e). Conversely, *CYP2E1, CYP3A4, IGFBP1, BAAT, SLCO1B3, KLF6* showed higher expression in central sub-clusters of both datasets (logFC < −0.4 for fresh and logFC < −1 for frozen) (Supplementary Fig. 3d, e).

For the LSECs, *VIM, EMP1, SPRY1, LMNA* showed particularly high expression in peri-portal subclusters (with the logFC > 0.9 for fresh and logFC > 2 for frozen) (Supplementary Fig. 4c, d) and *CLEC4G, STAB1, CLEC4M, OIT3, CTSD, LYVE1, CTSL, CD14* were significantly higher expressed in peri-central clusters (Supplementary Fig. 4c, d) in both fresh and frozen tissue datasets. Cell-cell interactions involving known genes related to angiocrine function used in Supplementary Fig. 14 have been obtained from[65–68].

**Zonation-specific protein expression validation using Human Protein Atlas.** Healthy liver immunostaining images (Supplementary Fig. 3f) were downloaded from the Human Protein Atlas[69] for expression information analysis of genes showing portal and central specific expression in our dataset.

### Statistics and reproducibility

Number of replicates is stated throughout the main text and in figure legends, as well as panels illustrating the study design (Fig. 1a; Fig. 2a). No statistical method was used to predetermine sample size, which was subject to human sample availability. No data were excluded from the analyses. The various statistical tests used are detailed throughout the main text, methods, and figure legends. A $p <= 0.05$ was considered statistically significant for the various statistical tests performed. The experiments were not randomized. The Investigators were not blinded to allocation during experiments and outcome assessment.

### Reporting summary

Further information on research design is available in the Nature Portfolio Reporting Summary linked to this article.

## Data availability

Raw and processed scRNA-seq and snRNA-seq data generated and used in this study have been deposited in ArrayExpress under accession E-MTAB-12594 and Mendeley (https://doi.org/10.17632/yp3txzw64c.1), respectively. Imaging data used in this study for lobular density investigation was deposited in https://doi.org/10.5281/zenodo.4772378. Source data are provided with this paper.

## Code availability

R notebooks and scripts used in this analysis can be found in https://github.com/tomasgomes/liver_regen and https://github.com/ReneHaensel/Liver_regen_cell_Atlas.

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

## Acknowledgements

We thank the Camp and Treutlein labs for helpful discussions, in particular C. Harmel for helpful discussions on 4i data analysis. We thank S. Pääbo, K. Köhler, B. Nickel, B. Schellbach, A. Weihmann, J. Kelso of Max Planck Institute for Evolutionary Anthropology for supporting this project. We also would like to acknowledge Markus Morawski from Paul Flechsig Institute of Brain Research at University Leipzig for sharing the slide scanner AxioScan Z1 (Carl Zeiss) and Karsten Winter from the Institute of Anatomy at University Leipzig for sharing the slide scanner Pannoramic Scan 2 (3DHISTECH). J.G.C. and B.T. are supported by grant number CZF2019-002440 from the Chan Zuckerberg Initiative DAF, an advised fund of the Silicon Valley Community Foundation. J.G.C. is supported by the European Research Council (Anthropoid-803441) and the Swiss National Science Foundation (Project Grant-310030_84795). B.T. is supported by the European Research Council (Organomics-758877, Braintime-874606), the Swiss National Science Foundation (Project Grant-310030_192604), and the National Center of Competence in Research Molecular Systems Engineering. T.G. is supported by an EMBO Long-Term Fellowship (ALTF 738-2019). M.D. is supported by the European Union through Horizon 2020 Research and Innovation Program under Grant No. 810645 and the European Union through the European Regional Development Fund Project No. MOBEC008. J.H. and D.S. are supported by the German Research Foundation (DFG, Germany) grant number: HA3091/14-1, HA3091/12-1 and SE 1694/4-1, SE 1694/5-1, respectively.

## Author contributions

A.B. generated the single-cell and single-nucleus data. T.G., A.B., Z.H., and M.D. analyzed the data. C.K. collected tissue samples and performed liver cell isolations. A.B. and T.S. performed single-nucleus suspension sorting using FACS with assistance from M. Santel. R. Hänsel and J.C.E. performed immunohistochemistry. R. Holtackers' and T.G. designed the 4i experiment. R. Holtackers' performed 4i sample preparation with assistance from M. Seimiya, as well as imaging acquisition and preprocessing. T.G. analyzed the 4i data, with supervision by P.W. D.S. was responsible for liver surgery, patient acquisition, clinical data, and CT/MRT analysis. T.D. was responsible for the PVE, CT/MRT imaging. M.B. and J.H. provided intellectual guidance for data analysis. A.B., T.G., M.B., J.H., G.D., B.T., and J.G.C. designed the study. A.B., T.G., G.D., B.T., and J.G.C. wrote the manuscript with input from all other authors. All authors read and approved the manuscript.

## Competing interests

The authors declare no competing interests.
