## [Peer Review File · Nature Communications]

REVIEWERS' COMMENTS

Reviewer #3 (Remarks to the Author):

The manuscript has been revised according to my comments and suggestions.

The changes in the revised manuscript will reduce the chance of misinterpretation of data. My main comments have been addressed.

However, I would like to urge the authors to again edit the manuscript and refrain from any statements that could be misunderstood as a concrete hypothesis that is tested. For example, on the top of page 7 the authors phrase the sentence: "To understand if these gene expression changes reflected alterations to the lobule spatial organization,....". It would be better not to raise the expectation that alterations in the lobule spatial organization have been investigated in depth (please see also comments from the other reviewer), it would be better to phrase it - gene expression changes MAY reflect....

Furthermore, please use abbreviations consistently throughout the manuscript. Hepatic Stellate cells - HSC.

Reviewer #4 (Remarks to the Author):

Please find below my comments concerning the manuscript entitled "Cell atlas of the regenerating human liver after portal vein embolization" by Agnieska Brazovskaja and colleagues.

The report describes single cell/nuclei mapping of human liver after embolization. The authors describe in details the impact of embolization on the different cell types composing the liver. Interestingly, they observed major change in zonation of hepatocytes and endothelial cells. Furthermore, they establish that fibroblasts are likely to play a key role in regenerative mechanisms. Finally, they apply multiplexing immunostaining to reinforce their observations.

This study represents an interesting resource that will be useful to a broad number of groups in the hepatology field. The analyses are very strong, and the authors have answered the reviewers' comments. The absence of mechanistic validations is not essential as the data presented provide important information.

Reviewer #3 (Remarks to the Author):

The manuscript has been revised according to my comments and suggestions.

The changes in the revised manuscript will reduce the chance of misinterpretation of data. My main comments have been addressed.

However, I would like to urge the authors to again edit the manuscript and refrain from any statements that could be misunderstood as a concrete hypothesis that is tested. For example, on the top of page 7 the authors phrase the sentence: "To understand if these gene expression changes reflected alterations to the lobule spatial organization,....". It would be better not to raise the expectation that alterations in the lobule spatial organization have been investigated in depth (please see also comments from the other reviewer), it would be better to phrase it - gene expression changes MAY reflect....

Furthermore, please use abbreviations consistently throughout the manuscript. Hepatic Stellate cells - HSC.

We would like to thank the reviewer for taking the time to review our work. We have now rephrased and adjusted our made observations and conclusions regarding inferences obtained with single-cell RNA-seq and other used techniques throughout our manuscript. We also checked and updated abbreviations for presented cell types, including Hepatic Stellate cells (HSC).

Reviewer #4 (Remarks to the Author):

Please find below my comments concerning the manuscript entitled "Cell atlas of the regenerating human liver after portal vein embolization" by Agnieszka Brazovskaja and colleagues.

The report describes single cell/nuclei mapping of human liver after embolization. The authors describe in details the impact of embolization on the different cell types composing the liver. Interestingly, they observed major change in zonation of hepatocytes and endothelial cells. Furthermore, they establish that fibroblasts are likely to play a key role in regenerative mechanisms. Finally, they apply multiplexing immunostaining to reinforce their observations.

This study represents an interesting resource that will be useful to a broad number of groups in the hepatology field. The analyses are very strong, and the authors have answered the reviewers' comments. The absence of mechanistic validations is not essential as the data presented provide important information.

We would like to thank the reviewer for taking the time to access our work, and for the positive feedback.